# Estimating heritability and genetic correlations from large health datasets in the absence of genetic data

Gengjie Jia [1], Yu Li [2], Hanxin Zhang [1,3], Ishanu Chattopadhyay[1], Anders Boeck Jensen[4], David R. Blair[5], Lea Davis [6], Peter N. Robinson [7], Torsten Dahlén[8], Søren Brunak[9], Mikael Benson [10], Gustaf Edgren [8], Nancy J. Cox[6], Xin Gao [2] & Andrey Rzhetsky [1,3,11]*

Typically, estimating genetic parameters, such as disease heritability and between-disease genetic correlations, demands large datasets containing all relevant phenotypic measures and detailed knowledge of family relationships or, alternatively, genotypic and phenotypic data for numerous unrelated individuals. Here, we suggest an alternative, efficient estimation approach through the construction of two disease metrics from large health datasets: temporal disease prevalence curves and low-dimensional disease embeddings. We present eleven thousand heritability estimates corresponding to five study types: twins, traditional family studies, health records-based family studies, single nucleotide polymorphisms, and polygenic risk scores. We also compute over six hundred thousand estimates of genetic, environmental and phenotypic correlations. Furthermore, we find that: (1) disease curve shapes cluster into five general patterns; (2) early-onset diseases tend to have lower prevalence than late-onset diseases (Spearman's $\rho = 0.32$, $p < 10^{-16}$); and (3) the disease onset age and heritability are negatively correlated ($\rho = -0.46$, $p < 10^{-16}$).

[1] Department of Medicine, Institute of Genomics and Systems Biology, University of Chicago, Chicago, IL 60637, USA. [2] Computational Bioscience Research Center, Computer, Electrical and Mathematical Sciences and Engineering Division, King Abdullah University of Science and Technology (KAUST), Thuwal 23955, Saudi Arabia. [3] Committee on Genomics, Genetics, and Systems Biology, University of Chicago, Chicago, IL 60637, USA. [4] Institute for Next Generation Healthcare, Department of Genetics and Genomic Sciences, Icahn School of Medicine at Mount Sinai, New York, NY 10029, USA. [5] Department of Pediatrics, University of California San Francisco, San Francisco, CA 94158, USA. [6] Division of Genetic Medicine, Vanderbilt University, Nashville, TN 37232, USA. [7] Jackson Laboratory for Genomic Medicine, Farmington, CT 06032, USA. [8] Department of Medical Epidemiology and Biostatistics, Karolinska Institutet, Stockholm 171 77, Sweden. [9] Novo Nordisk Foundation Center for Protein Research, Faculty of Health and Medical Sciences, University of Copenhagen, Copenhagen 1017, Denmark. [10] Centre for Individualized Medicine, Department of Pediatrics, Linkoping University, Linkoping 58183, Sweden. [11] Department of Human Genetics, University of Chicago, Chicago, IL 60637, USA. *email: andrey.rzhetsky@uchicago.edu

Disease manifestation patterns across populations are informative in at least two ways: (1) each disease has a unique prevalence profile across a patient's age and sex, and (2) diseases co-occur in nonrandom sequences. Firstly, geneticists and physicians have long been aware of the differences between early- and late-onset forms of the same disease. Earlier-onset disease subtypes have been typically associated with severer symptoms and higher heritability[1,2]. However, this late-versus-early-onset logic does not generalize well across diseases—with notable exceptions. Huntington's disease, for example, has a relatively late onset, typically, during the third or fourth decade of a patient's life, but it is a Mendelian, highly heritable condition. Secondly, physicians appreciate and occasionally use the information from a patient's chronological disease sequence to assist in disease diagnosis and management[3]. Given the richness of information contained in these disease trend and comorbidity patterns, we hypothesized that they could be useful in dissecting the genetic and environmental determinants of pathogenesis.

Traditionally, there are three main approaches to estimate quantitative genetic parameters like heritability and genetic correlations, all of which require the same two inputs: (1) genetic information (e.g., relatedness or genetic variants), and (2) phenotypic information (the affected or unaffected status with respect to one or more diseases).

The simplest and most intuitive way of computing these quantities involves a comparison of disease pattern concordance among monozygotic and dizygotic twins[4,5]. Monozygotic twins are genetically identical and dizygotic twins share, on average, half of their genetic polymorphisms. Therefore, it is relatively easy to mathematically partition the genetic and shared environmental contributions to disease phenotypes.

A slightly more complex version of the same approach involves an analysis of the overall nuclear family phenotypic variance of parents and children[6]. Because parents are typically genetically unrelated, a parent and a child share with each other half of their genetic variants, as do siblings. One can mathematically subdivide the overall phenotypic variance into several components (genetic, individual environment, shared sibling environment, and shared couple environment).

Building on more modern technology, an orthogonal approach to estimating heritability and genetic correlations utilizes genome-wide association studies (GWASs) outputs. In this approach, numerous unrelated individuals are compared in terms of their single-nucleotide polymorphisms (SNPs). The SNP-heritability is estimated as a proportion of the overall phenotypic variance explained by the common genetic polymorphisms of the affected individuals[7,8]. In particular, when estimates are computed using effect-size summation over SNPs, we refer to this version of heritability estimate as based on a polygenic risk score (PRS)[9,10].

The recent increase of both the importance and the availability of electronic health records (EHRs) has allowed us to scale family based analyses to millions of people[11,12]. However, methodologically, the procedure has not evolved much since the first family studies were performed. As a rule, EHRs are maintained in order to facilitate patient billing rather than academic research, and therefore, they are generally incomplete and biased[13]. However, this does not diminish their overall utility for making accurate inferences about clinical phenotypes in large populations. For example, administrative data has been successfully used to study asthma[14], autism[15], brain metastases[16], colonoscopy findings[17,18], colorectal and breast cancers[19], depression[20], glomerular filtration rate[21], polypectomy[22], and rheumatoid arthritis[23] in various populations. Furthermore, statistical epidemiologists routinely analyze insurance data to test for potential causal relationships between environmental factors and human maladies—for example, the effects of psychiatric pharmaceuticals on suicide rates[24].

The key to these types of analysis is to carefully examine how missing data and biases may affect the intended conclusions of the research and, if required, how to introduce appropriate, bias-neutralizing corrections[13].

The accumulating legacy estimates of genetic parameters, such as heritability and genetic correlations, pave way for the fourth approach that we are proposing here. Leveraging national EHR databases from the United States, Denmark and Sweden, we show that disease-specific statistics can be used to estimate heritability ($h^2$), inter-disease genetic/environmental/phenotypic correlations (corr) with an accuracy comparable to traditional clinical studies. These added estimates lead us to the findings that disease onset age is positively correlated with disease prevalence, but is negatively correlated with disease heritability.

## Results

**Defining and computing disease prevalence curves.** To capture the distribution of disease prevalence across age and sex, we computed disease prevalence curves by dividing the total number of disease codes (ICD codes) within each age-sex stratum by the number of enrolled patients matching these demographics (see Methods part 1 for the precise definitions). A disease prevalence curve's shape reflects the multiplicity of age-specific landmark events in a patient's life, ranging from health-neutral medical checkups (which can nevertheless reveal underlying conditions), to age-specific hormonal changes (e.g., puberty, pregnancy, or menopause), and to traumas and infections that may also correlate with age. Despite the existence of some country-specific variations (for example, in bipolar disorder, rheumatoid arthritis, and depression, see Fig. 1a), the curve shapes are rather consistent across countries for a large set of diseases, (see autism and gastrointestinal infection curves, and Supplementary Fig. 1 for selected examples, which were first discovered using the US[25] and Danish[26] cohorts, and then validated using the Swedish data[27,28]).

To further investigate shape-of-curve similarity, we defined a symmetric distance measure (see Methods part 2 for analytical details; Fig. 1b showing the full dissimilarity matrix). Our comparison across the whole disease spectrum and two countries (US and Denmark) identified five clusters (see Supplementary Fig. 2 for model selection results and Supplementary Data 7 for the complete list of over 500 studied diseases), and they have very distinct disease category, sex, and country compositions (Fig. 1c). For example, the smaller Clusters 4 and 5 primarily comprise neoplastic and developmental diseases, respectively. In these two clusters, the proportions of Denmark-derived disease curves are larger than US-derived ones. In contrast, US-derived curves are more common in Cluster 3. These clusters correspond to distinct shapes of curves: Cluster 1 corresponds to L-shaped early-onset conditions; Clusters 2 and 4 include reversed L-shaped curves (the former being early but slow rising, while the latter being later- but steeper-rising); Cluster 3 is the only multi-modal curve shape type; and Cluster 5 presents a skewed bell shape with a less heavy right tail than that of Cluster 1. For each cluster, we show a few representative disease curve alignments across different categories (indicated below in brackets). For example, in Cluster 1, parasitic infection aligns with an array of noninfectious diseases, including neurofibromatosis (hereditary and neoplastic), tympanic membrane disorders (otic), osteogenesis imperfecta (hereditary and musculoskeletal), and congenital eye anomaly (developmental). To the best of our knowledge, disease curves, standardized across age and sex, have never previously been systematically compared. The discovered resemblance, which can be of great interest to researchers and physicians, likely reflects a combination of shared factors in genetics (e.g., among autism,

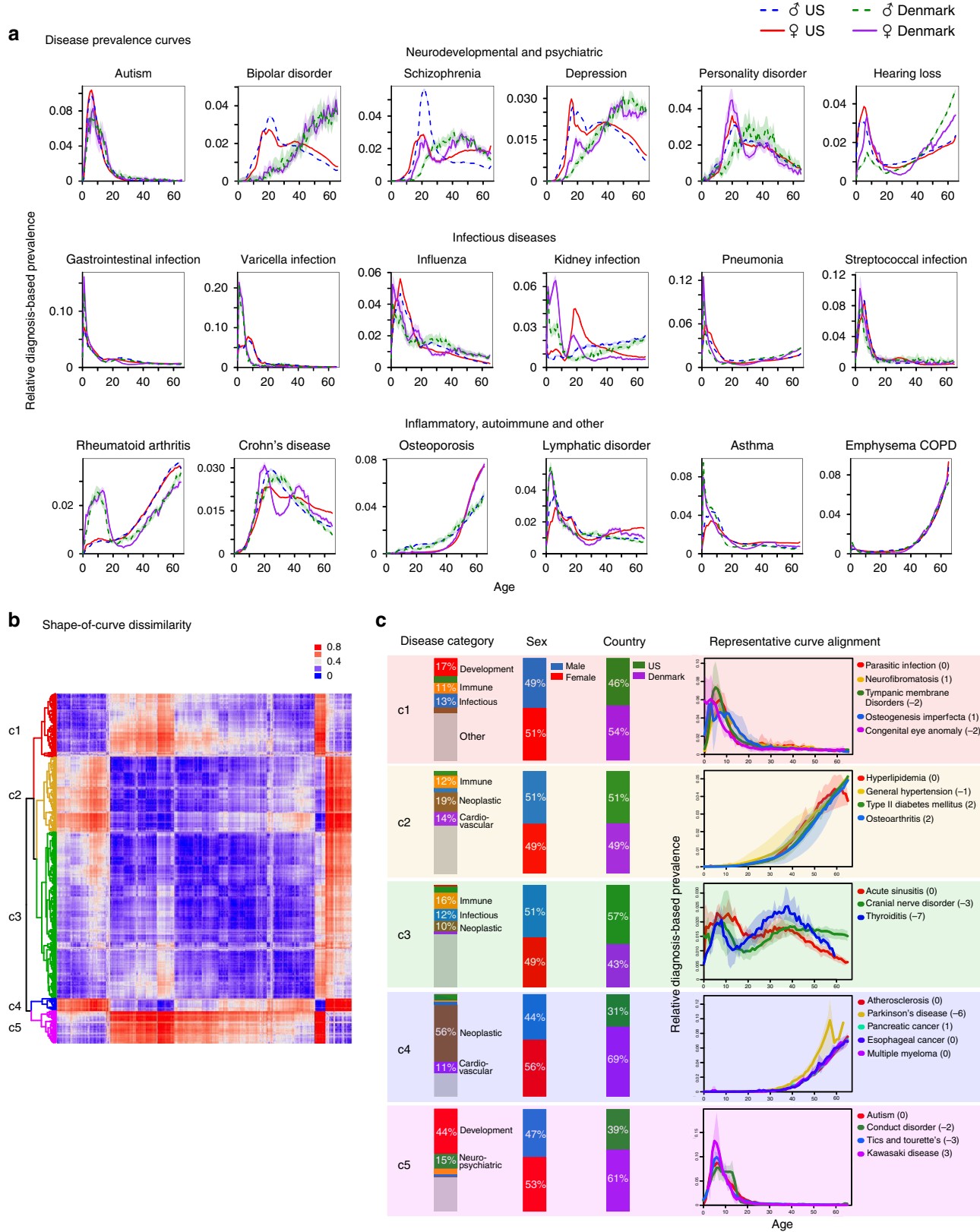

conduct disorder, tics, and Tourette syndrome[29]), environmental exposures, and developmental triggers (onset of puberty or menopause), or even direct causal links.

**Defining and computing disease embeddings**. To further augment disease similarity descriptors for this imputation procedure,

we implemented a disease embedding approach[30–32], inspired by natural language processing. To construct an embedding, we used a neural network as the mathematical representation of a word's underlying semantics, given its surrounding words (Methods part 3). The intuition behind this method is that semantically-similar words likely share contexts and would thus be encoded by similar

**Fig. 1 Disease prevalence curves fall into five major shape clusters. a** Representative disease prevalence curves for neurodevelopmental, psychiatric, infectious, inflammatory, autoimmune, and some miscellaneous diseases; we show disease names at the top of the corresponding plot. A curve's x-axis corresponds to the age of diagnoses (not necessarily the first one in the patient's recorded health trajectory), and the y-axis denotes the relative prevalence of each diagnosis in the corresponding age and sex group. For ease of comparison across countries, we re-normalized each curve to sum to 1. We computed the curves for two countries: the US and Denmark. US male-specific curves are depicted with blue-dotted lines and female-specific ones with red solid lines. As for their Danish counterparts, male-specific curves are shown with green-dotted lines and female-specific ones with purple solid lines. Each curve is supplied with a 99% confidence interval (in transparent colors). We find that some disease curves are consistent across countries and sexes (e.g., autism and gastrointestinal infection), while others vary by country only (e.g., bipolar disorder and rheumatoid arthritis), and still others vary by sex only (e.g., osteoporosis and Crohn's disease). **b** A distance matrix, shown as a heatmap, represents the shape dissimilarity between curves measured via the Jensen-Shannon divergence (Methods part 2). We applied a hierarchical clustering algorithm and elbow model selection to arrive at a five-cluster classification of curve shapes; the five clusters, $c1$–$c5$, are shown in red, yellow, green, blue, and purple, respectively. **c** At the left side of the plate, three columns of stacked bar charts summarize the compositions of each cluster in terms of disease category, sex, and country. At the right side of the plate, we show the optimal curve alignments (after relative shifts along the x-axis) of several representative diseases from each cluster. For each disease, we computed variations across four prevalence curve instances (two countries by two sexes), showing the curve mean and variation as a solid line and a shaded same-color area, respectively. The optimal relative shifts for the alignment are written as bracketed integer numbers (in years) after each disease name.

vectors of continuous parameters learned by a neural network. By analogy, chronologically ordered diseases in a patient's health record are "words," while the entire historical record becomes a "sentence." We employed over 151 million unique patient histories to compute the embedding for over 500 major diseases within a 20-dimensional continuous space, in which each disease is represented by a 20-dimensional vector (see Fig. 2 for snapshots of 3-dimensional projections of the embedding). To make our choice of dimensionality for the embedding space, we were driven by the following considerations: (1) the space dimensionality should be much smaller than the "vocabulary" size (over 500 in our case), but also be reasonably large enough to ensure adequate predictive power, and (2) the disease embedding with 20 latent dimensions should generate a reasonable nosology, as judged by physicians in our team. We further compiled a collection of published parameter estimates, including 1146 $h^2$ estimates and 1947 various $corr$ estimates (see Supplementary Data 1 and 2, respectively).

**Building estimators from disease descriptors.** Disease prevalence curves and disease embeddings derived from the US dataset were used as disease-specific descriptors for modeling. The modeling features also included specifications about predicted estimates (data type and mathematical model used), basic information about the investigated cohorts (country of origin and sex), and disease characteristics (category of biological systems that the disease belongs to, and the onset age). A detailed description of disease features used in the model can be found in Methods part 4. Equipped with various descriptors for individual diseases, disease–disease similarity measures, and large collections of legacy heritability and inter-disease correlation estimates, we proceeded to estimate missing genetic parameters across the whole pathology spectrum (see Fig. 3a for the outline of our modeling framework, and Fig. 3b–d for full collection of results). We tested a battery of modeling approaches, out of which a gradient boosting regression performed the best[33–35] (see Table 1 and Methods part 7 for details). As a measure of estimate quality, we used Pearson's correlation between imputed values and "actual" parameter values, training the ensemble regression model on 80% of the data and testing it on a held-out 20%. To ensure that results were not biased by a single lucky data split, we repeated this computation for 1000 randomly partitioned datasets.

Contour plots in Fig. 3b, c show the joint distributions of model predicted and previously published estimates: in the case of $h^2$, the density peaked around (0, 0) and (0.4, 0.4), indicating denser collocations of published and predicted estimates there; while as for $corr$, the estimates exhibited a unimodal distribution with a

peak close to (0.05, 0.05). The slopes of both linear regressions were close to 1, with negligible intercepts, indicating that our estimates were nearly perfectly unbiased. The correlations between our predictions and the corresponding legacy estimates had means of $0.870 \pm 0.001$ (95% confidence interval computed based on 1000 replicates) for $h^2$ and $0.874 \pm 0.001$ for $corr$ (Fig. 3d). As shown in Supplementary Table 1, over 40% of the useful information is from disease prevalence curves, over 30% from disease embeddings, and the rest mostly attributable to data types and mathematical models used in the relevant published studies. A detailed breakdown of the contribution of the 20 embedding factors shows that all the 20 factors contribute nearly equally.

To evaluate if our estimators were reasonably accurate, we computed a measure of agreement among previously published $h^2$ estimates. For this comparison, we used only published, independent estimates, matched both by data type and estimation methodology (if there were more than two estimates of the same type, we used all of them, generating all possible comparison pairs). We obtained 205 pairs of estimates in total and computed a Pearson's correlation value of 0.51, Student's $t$ test $p = 3.5 \times 10^{-15}$ (Supplementary Fig. 3a); agreement among these past estimates was much lower than what we observed for our estimates. Furthermore, the comparison between our estimates and very recently published sets of estimates[12] (which were used in neither training nor validation in our analysis) showed a significantly higher concordance between the two sets of estimates than legacy data (Pearson's correlation 0.71, Student's $t$ test $p = 4.8 \times 10^{-22}$, the number of estimates for comparison = 136; see Supplementary Fig. 3b and Supplementary Data 3 for comparison details). Similarly, to assess the accuracy of correlation estimates, we were able to identify an additional, independent dataset of genetic correlations[36] and reserved it exclusively for testing purposes. This test dataset was generated in context of GWASs and using a linkage disequilibrium score (LDSC) regression, we compared our predictions for the same data type and mathematical method. This confirmed a significantly high concordance (Pearson's correlation = 0.73, Student's $t$ test $p = 1.7 \times 10^{-14}$, the number of estimates for comparison = 80; please see Supplementary Fig. 3d and Supplementary Data 5 for comparison details). We therefore cautiously claim that our estimates are at least as good as those computed with traditional methods.

The addition of numerous genetic parameter estimates helped to statistically empower our downstream findings.

**Properties of disease heritability estimates.** Our initial analysis of US medical data generated a curious finding: The apparent overabundance of diseases with early onset. The distribution over

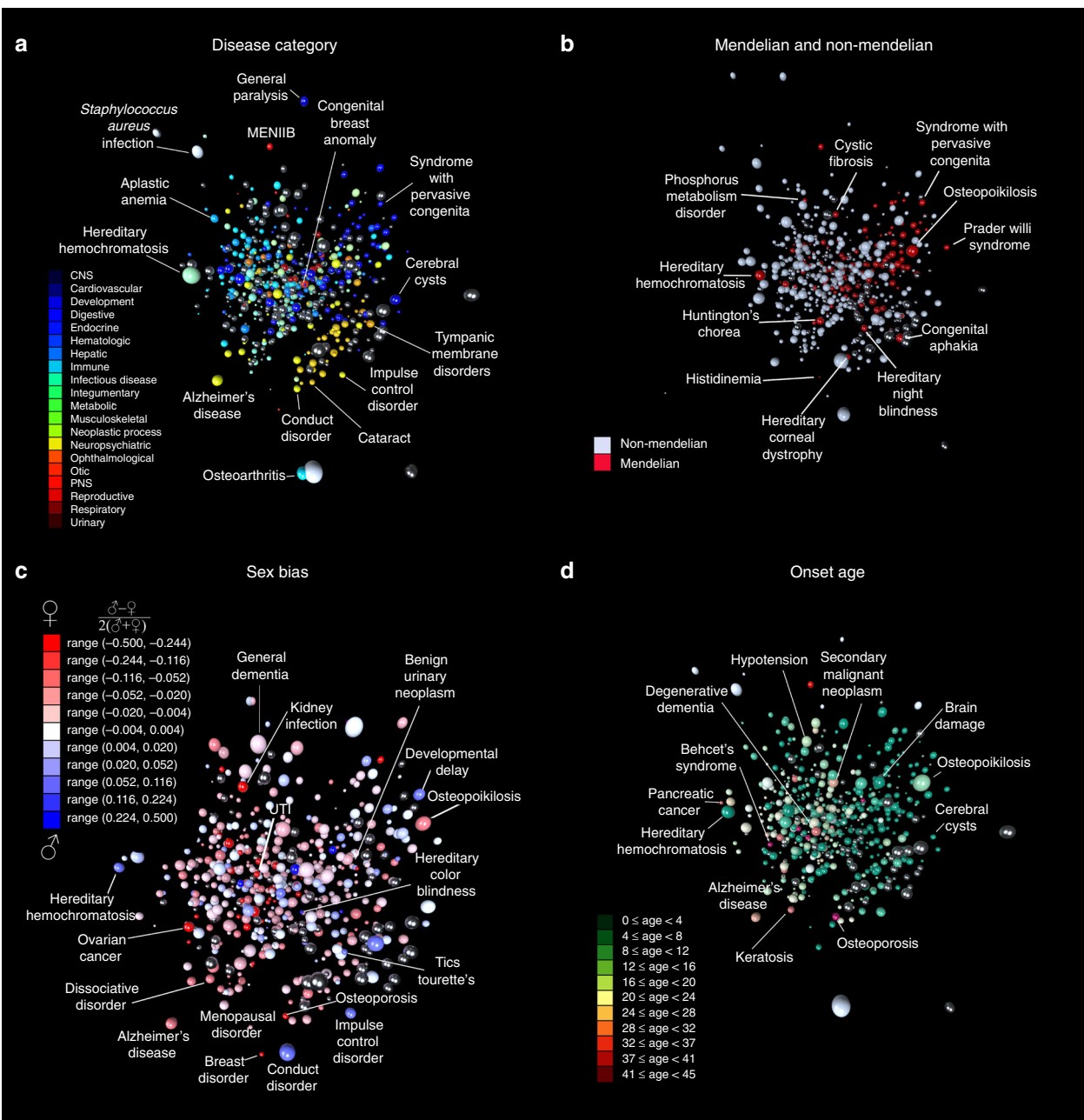

**Fig. 2 An embedding disease mapping into metric space positions, with related diseases close to each other.** Diseases can be mapped to points in low-dimensional metric space (so-called "disease embedding"). See the three-dimensional projections of our 20-dimensional embedding in (**a**)–(**d**) in this figure, where similar diseases are closer to each other in metric space than dissimilar ones. This 20-dimensional disease embedding turned out to be extremely useful in this study for estimating population-genetics parameters for individual diseases. **a** We projected the 20-dimensional disease embedding vectors of over 500 diseases into 3-dimensional space for ease of visualization, using the t-SNE algorithm[52]. We color-coded the spheres representing the diseases by each corresponding disease category. Plate **b** shows Mendelian vs. non-Mendelian disease distribution. Plate **c** shows disease-specific sex bias (defined in such a way that it is 0 for diseases that are equally frequent in males and females, −0.5 for diseases that occur only in females, and +0.5 for those occurring only in males). Plate **d** shows diseases color-coded in accordance with their onset ages, where green colors indicate early-onset childhood diseases, and warmer colors point to later-onset diseases.

the mean age of first disease diagnosis is approximately bell-shaped but skewed towards younger ages, with onset age mode around 42 years over all diseases (Fig. 3e). The total number of disease assignments is positively correlated with disease onset age (see Methods part 1 for the precise definition; Spearman's correlation is $\rho = 0.32$, $p < 10^{-16}$ computed using algorithm AS 89 (refs. [37,38]), as shown in Fig. 3f), and individual shape clusters all agree on the positive correlation (Methods part 5, Fig. 3g, and

Supplementary Table 2). This observation suggests that early-onset diseases outnumber late-onset diseases in the human population, and the former tend to have lower prevalence. Possible explanations for this could be associated with: (1) current clinical practice, such as routine newborn screening and monitoring, generating an overabundance of early-life health observations; (2) a tendency for conditions with substantial genetic etiology to have an earlier onset age while simultaneously being

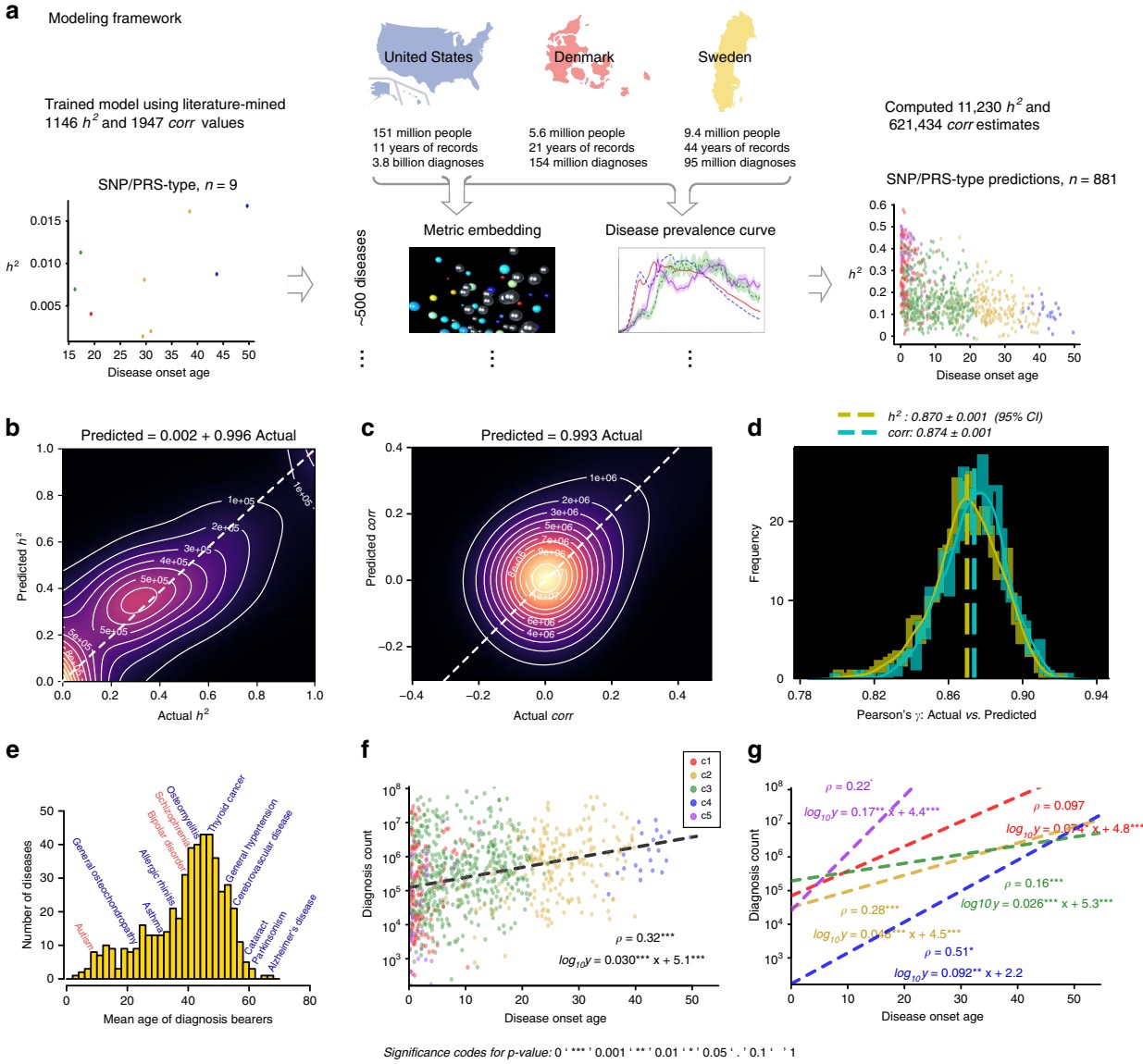

**Fig. 3 Estimating population-genetics parameters for hundreds of diseases and thousands of disease pairs.** Here, $h^2$ denotes heritability, and *corr* is a correlation between a disease pair which can be genetic, environmental, or phenotypic. **a** A workflow explains the key steps of our model development. We used three national-scale health registries, representing the United States, Denmark, and Sweden, which comprised 3.8 billion, 154 million, and 95 million disease diagnoses, respectively. We computed curves reflecting disease prevalence by age and sex (disease prevalence curves) and derived a metric mapping (disease embedding in metric space) for the whole disease spectrum. We used these two complementary representations to estimate hundreds of thousands of disease-specific parameters. We then validated the accuracy of our model's predictions by benchmarking them against previously-published ("actual") estimates that were not used in model training. Plates **b** and **c** show kernel density estimation plots we computed from 1000 random 4:1 splits of data (4/5 for training and 1/5 for testing). We used these plots to visualize the joint distribution of the actual data for testing and model-predicted values. The linear fit slopes between the actual and predicted values are 0.996 for $h^2$ and 0.993 for *corr*, indicating nearly perfectly unbiased estimations. **d** The distributions of Pearson's correlations between the actual and predicted values have mean values of 0.870 for $h^2$ and 0.874 for *corr*. **e** A distribution of the mean age of disease-specific diagnosis bearers. The median of the mean ages over all diseases is around 42 years, and specifically, the mean ages of autism, bipolar disorder, and schizophrenia that appeared in the US data are 9, 40, and 41, respectively. **f** There is a significant positive correlation between disease onset age and diagnosis count in the US data, suggesting there are less-than-expected, rare, late-onset diseases. **g** The relationship also holds for each of the five disease clusters. For individual clusters (*c1*–*c5*), we show the best linear approximation, regression coefficients (*p* values were computed using Student's *t* test), and Spearman's correlation $\rho$ (*p* values were computed using algorithm AS 89), color-coded by the shape cluster. Superscript asterisks indicate significance level of the estimates being different from 0.

driven to lower prevalence through negative selection; and (3) a historical bias in biomedical discovery since early-onset diseases were easier to document and categorize.

Because hypotheses (1) and (3) mentioned above cannot be tested with the data currently available to us, we focused first on hypothesis (2) and sought evidence of a systemically increased genetic load among early-onset diseases, using a narrow-sense heritability. The relevant legacy estimates were sparse (Fig. 4a and Supplementary Fig. 4a plot the twin/family and SNP/PRS-type estimates, respectively, against disease onset age). This study's imputation analysis added about 800 estimates to twin/family and SNP/PRS-type heritability estimates. The overall linear relationship between onset age and heritability was significantly, negatively sloped (Fig. 4a, b and Supplementary Fig. 4b).

**Table 1 Performance comparison of different modeling algorithms.**

| Modeling algorithms | 95% CI of Pearson's $\gamma$ for $h^2$ prediction | 95% CI of Pearson's $\gamma$ for *corr* prediction |
|---|---|---|
| Kernel Ridge regression | 0.837 ± 0.001 | 0.867 ± 0.001 |
| Lasso | 0.823 ± 0.002 | 0.793 ± 0.001 |
| Huber regression | 0.827 ± 0.001. | 0.787 ± 0.001 |
| Ridge regression | 0.826 ± 0.001 | 0.795 ± 0.001 |
| Random forest | 0.854 ± 0.001. | 0.856 ± 0.001 |
| Support vector regression | 0.856 ± 0.001 | 0.808 ± 0.001 |
| AdaBoost random forest | 0.858 ± 0.001 | 0.858 ± 0.001 |
| Gradient boosting regression | 0.870 ± 0.001 | 0.874 ± 0.001. |

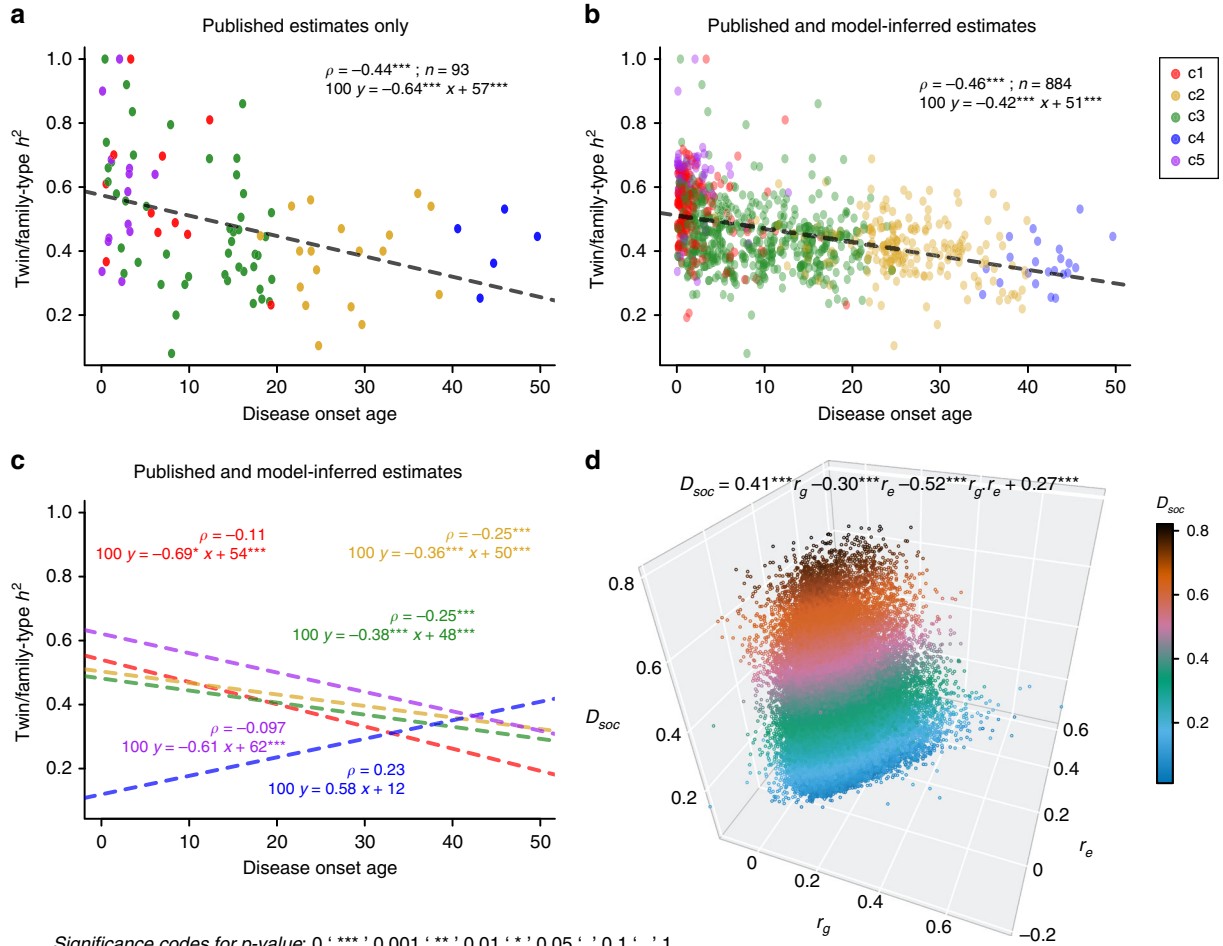

*Significance codes for p-value*: 0 ' *** ' 0.001 ' ** ' 0.01 ' * ' 0.05 '. ' 0.1 ' ' 1

**Fig. 4 Analyses empowered by our estimates of heritability ($h^2$), and genetic and environmental correlations ($r_g$ and $r_e$).** Plate **a** includes analyses solely based on the previously published estimates of twin/family-type $h^2$, suggesting a significantly negative correlation between disease onset age and heritability. **b** Our estimator substantially enriched the collection of twin/family-type $h^2$ estimates, filling in numerous missing estimates for under-studied diseases. When we analyzed disease prevalence curves jointly, we found a significantly negative correlation between disease onset age and $h^2$, which also holds for $h^2$ estimates based on other data types, such as SNP/PRS-type (Supplementary Fig. 4b). **c** We performed the same analysis for diseases within each of the five curve shape clusters, also confirming the significantly negative correlations for shape Clusters 1–3. In the smaller Clusters 4 and 5, the correlations were not significant (Methods part 5). **d** To understand the relationship between a disease pair's dissimilarity of disease prevalence curves ($D_{soc}$), and the $r_g$ and $r_e$ for the same disease pair, we performed a regression analysis, expressing $D_{soc}$ as a function of $r_g$, $r_e$, and an interaction term $r_g \cdot r_e$ ($p$ values were computed using Student's $t$ test, see Methods part 6). The corresponding regression coefficients turned out to be 0.41, −0.30, and −0.52, respectively. This regression analysis suggests the following: When two diseases have only high genetic correlation, their prevalence curves are likely to be very different; if only environmental correlation is high, the prevalence curves tend to be much more similar. However, disease prevalence curves are most similar when both environmental and genetic correlations between the two diseases are high. The included disease pairs across all categories are represented as hundreds of thousands of data points in the plot and they are colored according to the $D_{soc}$ values. We also repeated the same $D_{soc}$ regression analysis with all disease pairs from distinct disease categories (see Supplementary Data 6 and Supplementary Fig. 5).

However, when examined individually, the five-disease prevalence curve shape clusters exhibited heterogeneous behavior (Methods part 5, Fig. 4c, and Supplementary Fig. 4c). The detailed results from this analysis are provided in Supplementary Table 3.

Second, we asked, what can disease prevalence curve similarity tell us about the interplay between the genetic and environmental causes[39,40] for two diseases? A simple way to interpret the way in which nature and nurture affect the temporally manifested pattern of two diseases, is to perform a regression of dissimilarity between shapes of curves, $D_{soc}$, and genetic ($r_g$), and environmental ($r_e$) correlations between two diseases (Methods part 6, Fig. 4d, and Supplementary Data 6). The best-fitting regression curve appeared to be given by equation $D_{soc} = 0.27 + 0.41 r_g − 0.30 r_e − 0.52 r_g r_e$, where all regression parameters were significantly different from zero (Student's $t$ test, the largest $p = 3.6 \times 10^{-8}$). What this equation conveyed to us can be summarized as follows: when two diseases have only high genetic correlation, their prevalence curves are likely to be very different; if only environmental correlation is high, the prevalence curves would be much more similar. However, disease prevalence curves are most similar when *both* environmental and genetic correlations between the two diseases are high.

## Discussion

In addition to rigorously testing the hypotheses central to this study (the correlation between age of onset and disease heritability), we formulated a number of conjectures that await confirmation elsewhere, such as the link between similar disease curve shapes and hypothetically similar disease etiology. This study is intended to stimulate discussion and new thinking about factors affecting disease curve shapes and disease embedding properties. We hope that an approach like the one suggested here can eventually help to elucidate pathogenic mechanisms of common, complex disease, where genetic predisposition interacts with specific environmental insults to produce common disease symptoms.

We aimed to impute genetic parameter values for diseases in a gender-, data type- and model-specific manner in the absence of genetic data. Our main hypothesis in this quest was that this goal could be achieved by leveraging country-scale disease comparison information (prevalence curves and embeddings introduced in this study) and proper mathematical modeling. This hypothesis appears to be supported by the data we present in this study.

To understand the contribution made by various predictive features to our estimator's quality, it was useful to compute the relative importance of the disease-specific features as compared to that of all features used in the analyses. Our computations show that curve- and embedding-related disease features contribute heavily to the quality of the estimate of $h^2$ with 44.6% and 36.8%, respectively, and 81.4% collectively. Therefore, our newly engineered disease comparison features provide an essential contribution to overall prediction quality. Note that while our training cohorts come from 23 countries, our feature importance analysis shows that the predictor about the country of cohort contributes only 3.7% to the overall prediction quality for $h^2$ values. In other words, disease heritability estimates appear largely universal, not significantly affected by variation across populations.

We were very selective when including diseases into our prediction set; we only attempted estimations for diseases that were reasonably covered in the training dataset in the first place. For instance, because the majority of previous studies about genetic correlation estimation were either based on EHR-inferred pedigree information combined with the ACE (additive genetics, common environment, and unique environment) model, or SNP-based combined with the LDSC regression model, we limited our prediction outputs to these two settings exclusively.

The power of our designed disease features (curves and embeddings) is rooted in their deep connection to the genetic and environmental etiology of human pathology. Disease curves are shaped by a complex superposition of genetic predispositions, human physiological milestones (such as hormonal changes), social norms and incentives (such as youth participation in athletic activities), and environmental influences (exposure to periodic infections, pollution, traumas, and medications). Disease embeddings capture a disease "synonymy" that is also highly dependent on cultural and environmental conditions.

The culture-specific variations that influence disease prevalence become especially clear when we think of infectious diseases. A vivid, if gruesome, example is associated with Kuru, a fatal disease endemic to the eastern New Guinea Highlands. After a long search for infectious agents, the disease was linked to prions transmitted between people via an act of ritual funeral cannibalism[41]. There are many other less exotic examples, involving tropical infections (Ebola, Marburg, yellow fewer), seasonal infections (stomach and seasonal influenza), and arthropod-borne diseases (Lime disease, trypanosomiasis, and sickle cell disease). Not limited to infectious diseases, for example, there are rare psychiatric conditions, such as Koro (irrational perception of imminent loss of genitalia) in Southeast Asia[42]. Even diseases that are widely shared by nearly all cultures still have culture-specific variations in symptomatology and onset timing[43], and even vary within same-country sub-cultures[44]. It would be really fascinating to perform a systematic comparison of disease curves across numerous cultures and countries. Unfortunately, such comparative analyses are not feasible yet due to the lack of the required pan-cultural and pan-ethnic data.

Epidemiological literature has devoted considerable attention to disease variation across sexes and disease onset timing by focusing on one disease at a time. Investigators have looked at early-onset schizophrenia[45], concluding that the disease-to-sex ratio does not help to distinguish properties of early- and late-onset phenotypes. Both phenotypes present similar symptoms, with some developmental variation; delusions are less complex in children and are reflective of childhood themes. As for asthma, researchers studied the sub-forms associated with its onset time, and concluded that the early disease onset is associated with parental histories of allergy and asthma, genetic predisposition, and early-life environmental stresses, such as maternal smoking in pregnancy[46]. Gender was reported to play an important role in asthma as well; in females, asthma appears to be predominantly adult onset rather than pediatric[47]. More generally, investigators suggested that, in asthma, disease onset age determines distinct disease sub-types in adults[48]. Our study, based on nation-scale datasets, complements these traditional approaches by systematically comparing a diverse set of maladies using the concept of disease curves. Unlike traditional studies, we looked at a broad variety of diseases and have suggested that seemingly unrelated maladies show starkly similar disease curves, which might suggest a partially shared etiology.

A disease curve documents statistically significant changes in a malady's prevalence over the average lifespan. From a curve's extrema and inflections, one can identify patient ages that correspond to apparently distinct disease types. Then, for a given sex and age, one can search for under- and over-represented events in the lives of millions of patients. Some of the environmental trigger events for selected diseases are known, such as puberty, trauma, changes in dietary habits, and an interaction with a pathogen-rich environment. Apparent disagreements in sex-specific curves for the same disease, such as the presence of a curve extrema in females that is absent in males, may point to

previously ignored or as-yet undiscovered causes affecting the health of the population. A disease curve allows researchers to focus on relatively narrow, age- and sex-specific factor subsets associated with a given disease. With this narrowed collection of candidate factors, one can further test for their statistical association with a disease in an independent population of patients.

The reader may wonder whether acute and chronic diseases have distinct properties in terms of genetic parameter estimation quality? To answer this question, we performed a comparison of those acute and chronic diseases seen in the test dataset (see Supplementary Data 4). We first computed absolute errors (the absolute difference between inferred and published values), and then used a Wilcoxon rank sum test to determine whether the error distribution seen in acute diseases is different from that seen in chronic diseases. This difference proved to be nonsignificant ($p = 0.18$, Supplementary Fig. 3c), suggesting that the model prediction accuracy for acute diseases was not different from that for chronic ones. Both very much benefit from the rich information contained in disease trends and comorbidity patterns, which makes dissecting the genetic and environmental determinants of their pathogenesis possible. This observation might also be understood in context of the realization that the distinction between chronic and acute diseases is artificial in many cases. For example, a hemorrhagic stroke is an acute disease that is preceded by a chronic worsening of cardiovascular health and weakening blood vessel walls, resulting in a catastrophic (acute) rupture of a blood vessel (stroke). We conjecture that the heritability of the chronic stage of vessel weakening should be very similar to the acute stroke outcome.

To summarize, we computed two types of disease metrics, covering all human nosology categories. These metrics enabled us to: (1) impute genetic parameters for hundreds of diseases and thousands of disease pairs; (2) systematically analyze the relationship between heritability and disease onset age; and (3) relate shape-of-curve dissimilarity to genetic and environmental correlations between diseases. In addition, we provide a searchable web resource including all sex- and country-specific disease prevalence curves for over 500 diseases (see the link to the resource in Data Availability).

## Methods

**Disease prevalence curve**. In analysis A, applied only to the US dataset, we counted only a single disease diagnosis code per patient per year. We looked at ages between 0 and 65 years, inclusive; this age limit was imposed by the US MarketScan enrollment composition. To normalize these raw counts by recorded patients, we divided this sum of unique (per patient, per age) disease occurrences per year in a given sex-and-age group by the total number of visible patients in that demographic group. (To give an example, imagine a hypothetical patient, visible in the data for 2 years. The first year of visibility in the data is at age 35, wherein she has three diagnosis codes of disease X, and in the second year of visibility, at age 36, she has seven disease X diagnosis codes. To compute the disease curve, we counted this patient once in females with X-disease, age 35, and once in females with X-disease, age 36. We normalized each number by the total number of female enrollees at the corresponding age, so this hypothetical patient was counted once among enrollees of age 35, and once among enrollees of age 36.) This analysis implies that we were estimating, for each point of the curve, the expected proportion of patients in the specific sex and age group who will carry the current disease diagnosis. To convert the curve into a probability distribution, we normalized the raw estimates to sum to 1.

We designed this analysis in order to infer disease onset age, defined as the maximum age among the 5% of the youngest patients carrying the disease (i.e., the age at which the inverse distribution function for the disease curve is 0.05).

In analysis B, we did not use enrollment data (it was not available to us for the Scandinavian datasets). Instead, we estimated the expected share of disease X in a given demographic group. In other words, for each disease, we computed the total number of disease diagnoses in the sex-and-age group and normalized it by the sum of all disease diagnosis codes in this group. We further re-normalized the disease curves to sum to 1 to enable the curve comparisons across countries. We applied this procedure repeatedly to compute curves for all the diseases recorded in the databases (see Supplementary Data 7 for the complete list) and for all the combinations of sex-and-country groups (see the searchable web database

https://gjia.shinyapps.io/disease_curves/). Two national-level electronic medical record datasets were employed as discovery cohorts: one from the Truven Health Analytics MarketScan Commercial Claims and Encounters Database in the United States for the years from 2003 to 2013 (ref. [25]), and the other from the Danish National Patient Registry covering the years from 1994 to 2014 (ref. [26]) (Fig. 1). For the purpose of validation, we introduced another independent dataset, the Swedish National Health Registry, which covers the entire Swedish population's inpatient visits between 1968 and 2011 (refs. [27,28]) (Supplementary Fig. 1).

Furthermore, we use the US dataset to show that: (1) it is well representative of the general US population (Supplementary Fig. 6); (2) the curve computation is robust to variation in modeling hyperparameters, such as the enrollment year (Supplementary Fig. 7); and (3) the curves are also robust in their general properties for early-onset conditions, when computed exclusively from the newborn subpopulation (Supplementary Fig. 8).

**Clustering disease prevalence curve shapes**. Clustering analysis was based on US and Denmark datasets, and we arrived at the five-cluster curve classification shown in Fig. 1b using the following steps:

1. For each curve pair, we computed and minimized its dissimilarity measure by shifting one curve with respect to the other along the x-axis. We have chosen the Jensen–Shannon divergence[49], introduced for measuring dissimilarity between two probability distributions. We shifted one curve with respect to the other one along the x-axis (a year at a time, trying −8 to +8-year shifts).
2. We repeated this computation for all possible sex-and-country-specific curve pairs, covering over 500 human maladies (see the heatmap representation of dissimilarity matrix in Fig. 1b).
3. Based on this matrix, we applied a hierarchical, clustering algorithm (a complete linkage method)[50] and computed a bottom-up cluster hierarchy.
4. Finally, to determine the optimal number of groups (clusters) with the elbow model selection method, we used the following steps:

   a. Assuming that the optimum cluster number is equal to $K$ ($K = 1, 2, …$), we measured the clustering compactness by total intra-cluster variation, defined as $\sum_{k=1}^{K} \sum_{x_i \in C_k} (x_i - \mu_k)^2$, where $x_i$ is a data point belonging to cluster $C_k$, and $\mu_k$ is the average of the data points in $C_k$.
   b. We computed the total intra-cluster variation repeatedly for different values of $K$ ranging from 1 to 25 and plotted these values against the total number of clusters (Supplementary Fig. 2).
   c. The location in the plot at which the decline of the total variation switches from fast to slow (the elbow location) is regarded as the indicator of the optimal cluster number. In this study, this optimal number is five (indicated by a dashed line in Supplementary Fig. 2).

**Disease embedding**. We used the word2vec algorithm[31,32], which was originally developed for natural language processing. In our implementation, we adjusted the algorithm in the following ways: (1) we used disease codes in place of natural language words; (2) we replaced sentences with a chronological sequence of patient-specific disease codes; and (3) we replaced the text corpus with a large collection of patient-specific diagnostic histories. In a typical word2vec output, words are mapped into a continuous semantic space, so that synonymous words are placed nearby. Therefore, we aimed to find a similarity-based disease representation. The formal goal of this algorithm is to build a real-valued vector representation for a disease $\omega$ in order to predict its context (co-occurring) diseases $\omega_{-}$ given the current disease and vice versa. Using the logarithm of likelihood, $\mathcal{L}$, the cost function can be expressed as

$$\text{cost} = -\mathcal{L} = -\sum_{\omega \in C} \log P(\omega | \omega_{-}), \qquad (1)$$

where $C$ represents our "corpus" of over 151 million unique patient histories for over 500 major diseases. We used this corpus to train a neural network model using the gensim package[30]. We used context size of eight disease codes.

As a result, each disease is represented by a 20-dimensional vector (see Fig. 2 for snapshots of 3-dimensional projections of the embedding). We justify our choice of dimensionality for embedding space by the following considerations: (1) the space dimensionality should be much smaller than the "vocabulary" size (over 500 disease types in our case), but also be reasonably large enough to ensure adequate predictive power, and (2) the disease embedding with 20 latent dimensions should generate a reasonable nosology, as judged by physicians in our team.

**Defining disease features for prediction**. Disease-specific features in our model included a set of derivatives from disease prevalence curves and disease embedding. Specifically, for heritability imputation (single-disease analysis), the curve-derived set comprised a collection of disease-specific counts, which we normalized to 1 (as defined in Analysis B of Methods part 1), between ages 0 and 65 as well as to cumulative counts. We defined the cumulative count for age $N$ as a sum of all normalized counts from age 0 until the age $N$, inclusively. The embedding-derived set included all 20 real-valued elements in the 20-dimensional embedding vector.

We supplemented these two sets of features with a "biological system" label (a set of 20 labels shown in Fig. 2a, plus the label "Other"), the gender bias, the carrier's mean age, and the disease onset age.

As for correlation imputation (two-disease analysis), because disease pairs were involved, we used the mean and difference values of the normalized counts, cumulative counts, and embedding elements of each pair. In essence, these difference values captured disease-disease dissimilarities involving the comparison of single-disease features, such as distances between prevalence curves and between embeddings. Extending the one-disease supplemental features mentioned above, we also introduced disease-disease dissimilarities in their assigned biological system, in the gender bias, in the mean carrier age, and in the disease onset age.

For both single- and two-disease analyses, we also included categorical features to differentiate our predicted estimates by data type used, mathematical model, and basic information about the investigated cohorts (patient gender and country of origin). We used five data type labels ("twin study," "family study," "family study using EHRs," "SNP-based study," and "PRS-based study," as categorical one-hot-encoded variables), and six distinct labels to account for difference in mathematical models from published estimates ("AE," "ACE," "PRS," "SOLAR," "GREML," and "LDSC").

All training datasets for heritability and correlation imputation are available at https://github.com/jiagengjie/Estimating-Genetic-Parameters.

**Analysis of disease onset age.** For sex-specific heritability $h^2$, through an extensive literature search, we collected 1146 $h^2$ estimates for 403 unique diseases, but only 155 estimates for 68 unique diseases were gender-specific. These data were then substantially enriched by a set of estimates obtained in this study. If multiple estimates were available for a given disease and gender, we combined the estimates using the inverse-variance weighting method.

For correlation analysis, to investigate associations between disease onset age and the two metrics (diagnosis count and heritability), we applied Spearman's $\rho$ statistic and computed their $p$-values using algorithm AS 89 (refs. [37,38]) for the identified five clusters, both jointly and individually.

We performed regression analyses to fit linear models between disease onset age and either disease prevalence or heritability. We used the Student's $t$ test to determine whether the slope and intercept estimates significantly differed from zero. These results are reported in Figs. 3f–g, 4a–c, Supplementary Tables 2 and 3.

**Analysis of shape-of-curve dissimilarity ($D_{soc}$).** Through an extensive literature search, we gathered 812 estimates of genetic correlation $r_g$ and environmental correlation $r_e$. We then used our imputation procedure to extend this set of estimates to an exhaustive set of pairwise comparisons over approximately 500 diseases in total.

In a similar fashion, we performed regression analysis in intra- and inter-category disease pairs to fit models explaining $D_{soc}$ in terms of estimates of $r_g$ and $r_e$. We determined the slope's significance and intercept estimates being different from zero via Student's $t$ test. We report these results in Fig. 4d, Supplementary Fig. 5 and Supplementary Data 6.

**Model.** For model training, we collected 1146 $h^2$ estimates and 1947 corr estimates from 234 individual publications (Supplementary Table 4 lists a few representative, large-scale studies along with their key features, and the complete data can be found in the upper rows of Supplementary Data 1 and 2). We experimented with a few predictive methodologies, including generalized linear models (Lasso, Huber regression, and ridge regression), kernel ridge regression, support vector regression, and ensemble methods (random forest, AdaBoost random forest, and gradient boosting regression). These algorithms all performed rather well, as evaluated on 1000 repeated runs (in each run, we randomly selected four-fifths of the data for training and one-fifth for validation, see Table 1). Gradient boosting regression performed the best (Table 1 and Fig. 3b–d) and is explained in more detail below.

Given a training dataset of known output and input pairs $\{y_i, \mathbf{X}_i\}_1^N$, the algorithm's goal is to obtain an approximation to the function $F(\mathbf{X})$ that maps $\mathbf{X}$ to $y$ (denoted $\widehat{F}(\mathbf{X})$), such that the expectation of a loss function $L(y, \widehat{F}(\mathbf{X}))$ is minimized. The gradient boosting regression model utilizes an ensemble of predictor regression trees[33], built in a forward, stage-wise fashion to minimize a differentiable squared-error function $(y - \widehat{F}(\mathbf{X}))^2$. The pseudo-code for this computation is as follows:

$$F_0(\mathbf{X}) = \bar{y}$$
$$\text{For } m = 1 \text{ to } M \text{ do}:$$
$$\tilde{y}_i = y_i - F_{m-1}(\mathbf{X}_i), i = 1, \cdots, N$$
$$(\rho_m, \boldsymbol{\alpha}_m) = \arg\min_{\rho, \boldsymbol{\alpha}} \sum_{i=1}^{N} [\tilde{y}_i - \rho h(\mathbf{X}_i; \boldsymbol{\alpha})]^2$$
$$F_m(\mathbf{X}) = F_{m-1}(\mathbf{X}) + \rho_m h(\mathbf{X}; \boldsymbol{\alpha}_m)$$
$$\text{end For}$$

where $h(\mathbf{X}; \boldsymbol{\alpha}_m)$ and $\boldsymbol{\alpha}_m$ denote the base learners (regression trees) and the vector of model parameters (split locations and means of tree terminals). The number of trees $M$ and the learning rate $\rho_m$ are model hyperparameters, which we tuned to 200 and 0.1, respectively. We started with a model containing only the constant function $F_0(\mathbf{X})$, and incrementally expanded it in the for-loop as shown above[51].

Ultimately, we deployed this model to obtain estimates of $h^2$ and corr, not only for the complete spectrum of diseases and two sexes, but also for various data types and modeling assumptions (see Supplementary Data 1 and 2 for the complete collection of estimates).

## Data availability

We have launched a searchable web application for researchers to explore and compare sex-and-country-stratified prevalence curves for over 500 diseases. https://gjia.shinyapps.io/disease_curves/.

The license of MarketScan databases is available to purchase by Federal, nonprofit, academic, pharmaceutical, and other researchers. Access to the data is contingent on completing a data use agreement and purchasing the needed license. More information about licensing the MarketScan databases can be found at https://www.ibm.com/us-en/marketplace/marketscan-research-databases.

Access to individual-level Denmark data is governed by Danish authorities, including the Danish Data Protection Agency, the Danish Health Data Authority, the Ethical Committee, and Statistics Denmark. Researchers at Danish research institutions must obtain the relevant approval and data before initiating relevant scientific projects. International researchers may gain data access if supervised by a Danish research institution that has needed approval and data access.

The study has been approved by the ethical review board in the Stockholm county (DNR 2018/2153-31). Data storage and access is compliant with local laws and regulations.

All other data contained in the article and in its supplementary information are available upon request.

## Code availability

All codes, which compared various modeling algorithms for heritability and correlation imputation, and thus generated the results shown in Table 1, are available at https://github.com/jiagengjie/Estimating-Genetic-Parameters.

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

## Acknowledgements

We are grateful to E. Gannon, R. Melamed, R. Mork, M. Rzhetsky, and E. Wachspress for comments on earlier versions of this manuscript, and to H. Sanayle for advising us on Autodesk Maya 2019 Python programming. This work was funded by the DARPA Big Mechanism program under ARO contract W911NF1410333, by National Institutes of Health grants R01HL122712, 1P50MH094267, and U01HL108634-01, by a gift from Liz and Kent Dauten, and by funding from King Abdullah University of Science and Technology (KAUST), under award number FCC/1/1976-18-01, FCC/1/1976-23-01, FCC/1/1976-25-01, FCC/1/1976-26-01, and FCS/1/4102-02-01. This research made use of the resources of the Supercomputing Laboratory at KAUST.

## Author contributions

G.J., I.C., and A.R. designed the study; G.J., I.C., and A.R. analyzed data; G.J. and A.R. wrote the manuscript; Y.L. and X.G. tested machine learning algorithms; H.Z. and D.R.B. provided mappings between ICD codes and disease names; A.B.J. and S.B. prepared the Danish dataset; T.D. and G.E. prepared the Swedish dataset; and L.D., P.N.R., M.B., and N.J.C. advised on biomedical interpretations for the results.

## Competing interests

The authors declare no competing interests.
