## [Peer Review File · Nature Communications]

Reviewers' Comments:

Reviewer #1:

Remarks to the Author:

Restatement of the paper:

This paper provides a method to impute disease heritability and disease-disease genetic and environmental correlations from EHR data trained against literature-derived estimates. They generate disease prevalence curves and disease similarity low dimension embedding, using electronic health record (EHR) data, as features, as well as gender, country, study type and methodology. They collect published 984 h^2 and 1947 correlation estimations as true values. Then they build a machine learning model. They validate model results by hold-out validation, and comparing to another independent dataset. They predict the heritability and correlation for new "disease-study type-methodology-gender" combinations using the model. With new estimations, they find the negative correlation between disease onset age and disease heritability, and the relationship among disease dissimilarity and genetic/environmental correlations.

Overall, I believe this is an exciting and novel use of EHR data that has great potential for impact across many fields. I have outlined some major and minor concerns below.

Major concerns:

1. Dataset bias.

The true values of h^2 are come from previous published studies, which include only 290 diseases. This suggests the training and evaluating processes are based only on part of the 500 diseases, and the data of the other part of diseases is never met by the model. Therefore, the genetic parameter estimation of those unseen diseases may not be valid. Also, only 51 diseases have female estimations, and 34 diseases have male estimations. The gender-specific heritability estimation may not be valid.

In other words, because the training and testing data is limited by results of previous studies, it is not a random sample of the real data, which may limit the predictive power of the model.

The major result of the paper, the negative correlation between disease onset age and heritability, is mainly supported by the predicted values, so this result may be lack of evidence if the prediction of the model is not valid.

2. Lack of validation.

The disease-disease genetic and environmental correlations are only validated using intra-dataset cross validation, but not validated using independent datasets.

3. Unfair comparison

When evaluating the model performance, the authors argue that the prediction of the model is reasonably accurate, by indicating the correlation of the predicted value and true value is 0.4 higher than the correlation between previous published, independent estimations. But the calculation of correlation between previous published, independent estimations includes data from distinct populations or gender, which may explain the large variation, while the correlation of the predicted value and true value is measured using perfectly paired data. Thus, the comparison is not fair, and this evidence does not support the accuracy of the model.

Minor concerns:

1. Insufficient methodological detail

a) The method of the disease low dimensional embedding is not clear enough. It would better to explain the embedding in a separate paragraph in Methods.

b) How disease prevalence curve and disease low dimensional embedding translated to features of the model is unknown. More detail is required in the methods.

c) How disease prevalence curves and published results are paired as features in the aspect of country? The country-specific prevalence curve is measured by US, Denmark and Sweden electronic health record (EHR) data, while the published results are come from 17 countries, including Australia, Japan and Vietnam. Were comparisons made between countries? Are the results just as accurate?

- d) How the hyperparameters of the model are tuned? It's not clear which part of the data is used for hyperparameter tuning. The authors should ensure that the hyperparameters were fit using ONLY the training data and not the testing data. Using cross-validation within the 80-percent hold out may solve this issue.
- e) The term "disease mentions" is not clear in the manuscript. It seems to refer to diagnosis codes, but the way it's written is ambiguous.
- f) There was limited to no discussion of acute versus chronic diseases. Is the same level of performance achieved for stroke as diabetes, for example? Why or why not?
- g) Was the uncertainty (or errors) in the estimates of prevalence used in the models as well? Or just the point estimates?

Reviewer #2:

Remarks to the Author:

NCOMMS-19-12796

Estimating Genetic Parameters in the Absence of Genetic Data from Country-scale Health Datasets

This paper by Jia and colleagues has two major contributions. First, using large health care datasets, the authors computed disease prevalence curves and low dimensional disease embeddings which revealed similarity and dissimilarity of temporal disease trajectories across the disease spectrum. Second, the authors showed that disease-specific features and disease-disease relationships can be used to imputed unseen disease heritability and genetic correlation measures with high accuracy. Overall I think this is an interesting paper with many novel ideas and I applaud the authors for putting together very large scale health care datasets across countries.

However, I'm disappointed about the writing of this paper. I think it was initially written as a short letter to target a different journal. As a result, introduction, results and discussion were lumped together; the background was not thoroughly reviewed and motivation wasn't clear; and the implications of the results were not discussed. More importantly, the methods section is too brief to provide enough details to replicate the analysis. With the level of methodological details presented it's difficult to assess the rigor of the methods. In addition, the authors wanted to show lots of analyses and results in this paper but the flow of the text was sort of poor, making it difficult to follow. One has to jump between the main text, methods and figure legends to figure out what has been done.

If possible, I would suggest the authors rewrite and resubmit the paper. The paper is interesting but needs to be expanded and polished to more clearly present the results and methods, and to add background introduction and discussion.

Reviewers' comments (in black) and proposed possible answers (in red):

Reviewer #1 (Remarks to the Author):

Restatement of the paper:

This paper provides a method to impute disease heritability and disease-disease genetic and environmental correlations from EHR data trained against literature-derived estimates. They generate disease prevalence curves and disease similarity low dimension embedding, using electronic health record (EHR) data, as features, as well as gender, country, study type and methodology. They collect published 984 h^2 and 1947 correlation estimations as true values. Then they build a machine learning model. They validate model results by hold-out validation, and comparing to another independent dataset. They predict the heritability and correlation for new “disease-study type-methodology-gender” combinations using the model. With new estimations, they find the negative correlation between disease onset age and disease heritability, and the relationship among disease dissimilarity and genetic/environmental correlations.

Overall, I believe this is an exciting and novel use of EHR data that has great potential for impact across many fields. I have outlined some major and minor concerns below.

Major concerns:

1. Dataset bias.

The true values of h^2 are come from previous published studies, which include only 290 diseases. This suggests the training and evaluating processes are based only on part of the 500 diseases, and the data of the other part of diseases is never met by the model. Therefore, the genetic parameter estimation of those unseen diseases may not be valid. Also, only 51 diseases have female estimations, and 34 diseases have male estimations. The gender-specific heritability estimation may not be valid.

In other words, because the training and testing data is limited by results of previous studies, it is not a random sample of the real data, which may limit the predictive power of the model.

We very much appreciate the reviewer’s constructive suggestions and would like to address the concern about possible dataset bias in the following three ways:

1) To construct the current training dataset, we searched (almost) all the major studies about heritability and correlations, collecting 984 h^2 estimates and 1947 *corr* estimates from 190 unique publications. To address the reviewer’s concerns and further enrich the training dataset, we iterated an additional search over the list of “missing” diseases in the current dataset, focused on one disease at a time, looked for disease mentions and their heritabilities in the PubMed database, and then read through all the relevant publications to confirm. In this way, we were able to identify additional 75 published h^2 estimates for 49 unique diseases. In addition, we augmented our dataset with 87 Mendelian diseases for which the broad-sense heritability can be assumed to be 1 (Visscher, Hill et al. 2008).

Altogether, we managed to increase the total number of published estimates from 984 to 1146, in which **403** unique diseases and **155** gender-specific estimates were covered in total.

Further, using the same model-building procedure as before, we randomly selected four-fifths of the data for training and one-fifth for validation, and repeated this process 1000 times. As a result, the cross-validation and independent tests validated the accuracy of our estimator. Accordingly, we have updated Figures 3B, 3D, 4A-C, Supplemental Figures S3, S4, Tables 1, 2, and Supplemental Table S2.

- 2) Regarding the question raised about the difficulty in using 403 measured diseases to estimate parameters for 493 diseases in total, we conjectured that our model should be able to impute missing values because it has access to country-scale disease comparison information, which is embedded in temporal disease prevalence curves and 20-dimensional disease-specific embedding vectors. This conjecture appears to be well-supported by our result discussed above. In addition, it is informative to observe that the relative importance of disease-specific features shows among all predictive features. Specifically, curve- and embedding-related disease features contribute towards the estimates of h^2 44.6 percent and 36.8 percent, respectively, and 81.4 percent collectively.
- 3) Furthermore, we double-checked that our predictive power was not “overstretched.” We were very selective in including diseases into our training and prediction sets: We only attempted the estimation for diseases that were reasonably covered in the training dataset in the first place. For instance, because the majority of previous studies on genetic correlation estimation were either based on EHRs-inferred pedigree information combined with the ACE (additive genetics, common environment, and unique environment) model, or SNP-based combined with the LDSC (linkage disequilibrium score regression) model, we limited our prediction outputs to these two settings exclusively.

We have added these details to the Discussion section.

The major result of the paper, the negative correlation between disease onset age and heritability, is mainly supported by the predicted values, so this result may be lack of evidence if the prediction of the model is not valid.

Thank you for the comment. The reported negative correlation is also strongly supported by published (legacy) estimates, if we exclude numerous new estimates produced in our study. As shown in the original Supplemental Figure S4A and Table S2, disease onset age was negatively correlated with heritability: Spearman’s ρ was -0.44 (p -value = 1.2×10^{-5}) and the regression coefficient was -0.64 (p -value = 1.2×10^{-4}). When we combined legacy estimates with new ones, the negative correlation still held for both twin/family-study-based (Figure 4B) and SNP/PRS-study-based heritability (Supplemental Figure S4B).

As the reviewer suggested that the negative correlation between disease onset age and heritability was one of our study’s main findings, we elaborated on this point in the revised manuscript and have moved the original Supplemental Figure S4A to the revised main Figure 4. Hopefully, it is now appropriately highlighted in the current revised version.

2. Lack of validation.

The disease-disease genetic and environmental correlations are only validated using intra-dataset cross validation, but not validated using independent datasets.

Fortunately, we were able to identify an additional, independent dataset of genetic correlations (Tylee, Sun et al. 2018) and reserved it exclusively for testing purposes (we used this dataset for neither training nor intra-dataset validation in our analysis). This test dataset was generated in context of genome-wide association studies and using a linkage disequilibrium score (LDSC) regression, we compared our predictions for the same data type and mathematical method. This confirmed a significantly high concordance (Pearson's correlation = 0.73, p -value = 1.7×10^{-14} ; please see Supplemental Figure S3D and Table S10 for comparison details).

We have added these details to the Results section.

3. Unfair comparison

When evaluating the model performance, the authors argue that the prediction of the model is reasonably accurate, by indicating the correlation of the predicted value and true value is 0.4 higher than the correlation between previous published, independent estimations. But the calculation of correlation between previous published, independent estimations includes data from distinct populations or gender, which may explain the large variation, while the correlation of the predicted value and true value is measured using perfectly paired data. Thus, the comparison is not fair, and this evidence does not support the accuracy of the model.

We should clarify that the assessment of our model prediction quality was based on both repeated cross-validations and additional tests on independent datasets. We agree that the matching approach in our settings was favorable to our method. Therefore, we made efforts to improve comparison fairness in the revision:

- 1) In preparation for the correlation analysis between published independent estimates, we actually matched not only their data types and mathematical models, but also the genders of studied cohorts. Therefore, the variation shall only come from distinct studies using different cohorts, but good estimations shall be robust against such variation and provide coherent estimates. We think, analogously, correlating our model predictions with the published ones is essentially to compare estimates from two independent studies that adopted different methods (our modeling approach and published conventional approaches).
- 2) If our newly-predicted and published values are considered to be paired unfairly too perfectly, then applying the exact same criteria as we used to match between the past estimates, we compared our model predictions against very recently published sets of new estimates (Lakhani et al., 2019, which were used in neither training nor validation in our analysis). These new estimates still show that the concordance between two sets of new estimates (Pearson's correlation 0.71, p -value = 4.8×10^{-22}) is improved by 0.2, when compared against Pearson's correlation value of 0.51, as seen between the past estimates

(shown in Supplemental Figures S3A-B). In the revised manuscript, we used this as the supporting evidence instead.

Minor concerns:

1. Insufficient methodological detail

a) The method of the disease low dimensional embedding is not clear enough. It would better to explain the embedding in a separate paragraph in Methods.

Thank you for the suggestion. We expanded the description as requested.

Disease embedding

We used the word2vec algorithm, which was originally developed for natural language processing. In our implementation, we adjusted the algorithm in the following ways: (1) We used disease codes in place of natural language words; (2) We replaced sentences with a chronological sequence of patient-specific disease codes, and; (3) We replaced the text corpus with a large collection of patient-specific diagnostic histories. In a typical word2vec output, words are mapped into a continuous “semantic” space, so that synonymous words are placed nearby. Therefore, we aimed to find similarity-based disease representation. The formal goal of this algorithm is to build a real-valued vector representation for a disease ω in order to predict its context (co-occurring) diseases ω_- given the current disease and vice versa. Using the logarithm of likelihood, \mathcal{L} , the cost function can be expressed as

$$\text{cost} = -\mathcal{L} = -\sum_{\omega \in \mathcal{C}} \log P(\omega | \omega_-),$$

where \mathcal{C} represents our “corpus” of over 121 million unique patient histories for over 500 major diseases. We used this corpus to train a neural network model using the gensim package.

As a result, each disease is represented by a 20-dimensional vector (see Figure 2 for snapshots of three-dimensional projections of the embedding). We justify our choice of dimensionality for embedding space by the following considerations: (1) The space dimensionality should be much smaller than the “vocabulary” size (over 500 in our case), but also be reasonably large enough to ensure adequate predictive power, and; (2) The disease embedding with 20 latent dimensions should generate a reasonable nosology, as judged by physicians in our team.

We have added the above paragraphs as a separate section in the Methods 3, in order to explain disease embedding’s technical details.

b) How disease prevalence curve and disease low dimensional embedding translated to features of the model is unknown. More detail is required in the methods.

This is a very good suggestion. We have expanded the description as requested.

Defining disease features for prediction

Disease-specific features in our model included a set of derivatives from disease prevalence curves and disease embedding. Specifically, for heritability imputation (single-disease analysis), the curve-derived set comprised a collection of disease-specific counts, which we normalized to 1 (as defined in Analysis B of Methods 1), between ages 0 and 65 as well as to cumulative counts. We defined the cumulative count for age N as a sum of all normalized counts from age 0 until the age N , inclusively. The embedding-derived set included all 20 real-valued elements in the 20-dimensional embedding vector. We supplemented these two sets of features with a “biological system” label (a set of 21 labels shown in Figure 2A, plus the label “Other”), the gender bias, the carrier’s mean age, and the disease onset age.

As for correlation imputation (two-disease analysis), because disease pairs were involved, we used the mean and difference values of the normalized counts, cumulative counts, and embedding elements of the interested pairs. In essence, these difference values captured disease-disease dissimilarities involving the comparison of single-disease features, such as distances between prevalence curves and between embeddings. Extending the one-disease supplemental features mentioned above, we also introduced disease-disease dissimilarities in a biological system, in the gender bias, in the mean carrier age, and in the disease onset age.

For both single- and two-disease analyses, we also included categorical features to differentiate our predicted estimates by data type used, mathematical model, and basic information about the investigated cohorts (patient gender and country of origin). We used five data type labels (“twin study,” “family study,” “family study using EHRs,” “SNP-based study,” and “PRS-based study,” as categorical one-hot-encoded variables), and six distinct labels to account for difference in mathematical models from published estimates (“AE,” “ACE,” “PRS,” “SOLAR,” “GREML,” and “LDSC”).

All training datasets for heritability and correlation imputation are available at <https://github.com/jiagengjie/Estimating-Genetic-Parameters>.

We have added the descriptions above to the Methods 4.

c) How disease prevalence curves and published results are paired as features in the aspect of country? The country-specific prevalence curve is measured by US, Denmark and Sweden electronic health record (EHR) data, while the published results are come from 17 countries, including Australia, Japan and Vietnam.

Out of the published 1,146 h^2 estimates, only 22 were from studies based on an Australian cohort, four were from Japan, and one was from Vietnam; thus, the total number is small. In

order to still be able to utilize these estimates, we used dynamic prevalence curves measured by the US cohort as their proxies.

Were comparisons made between countries? Are the results just as accurate?

Yes, the results were still accurate, regardless of the country of analysis; country information has been proven not to be important in h^2 prediction, only 3.7 percent (feature importance). In addition, to predict heritability for conditions that were never measured, *e.g.*, diseases, genders, data types, and estimation methods, we only computed for the countries that were reasonably covered in the training dataset (*e.g.*, US and Sweden). In the case of the countries that were not considerably represented in the training datasets, we only output model-simulated values for the exact same condition used for model training, in order to assess our model's performance.

We have added these details to the Discussion section.

d) How the hyperparameters of the model are tuned? It's not clear which part of the data is used for hyperparameter tuning. The authors should ensure that the hyperparameters were fit using ONLY the training data and not the testing data. Using cross-validation within the 80-percent hold out may solve this issue.

As the reviewer suggested, we followed the standard machine learning procedure of hyperparameter tuning and testing, ensuring that the testing dataset was hidden from the hyperparameter tuning. More specifically, we applied a grid search approach, exhaustively searching through a set of those hyperparameters predefined by the machine learning expert. We divided the whole dataset into training (80 percent) and testing (20 percent) datasets. Within the training dataset, we performed five-fold cross-validation (CV) to determine the best hyperparameters for each regression model. Then, after fixing the hyperparameters based on our CV result, we trained the model's parameter using all the training data, then evaluated each regressor's performance on the hidden testing dataset. To ensure the robustness of our conclusions, we repeated the above procedures 1,000 times, which resulted in the performance comparison of different modeling algorithms shown in Table 1.

e) The term "disease mentions" is not clear in the manuscript. It seems to refer to diagnosis codes, but the way it's written is ambiguous.

As requested, the term "disease mentions" was replaced with "disease diagnosis codes."

f) There was limited to no discussion of acute versus chronic diseases. Is the same level of performance achieved for stroke as diabetes, for example? Why or why not?

Thank you for the thoughtful question. We checked a few examples of acute and chronic diseases, and confirmed that the model performance appeared to be consistent across disease types (we added a discussion of this observation to the revised text). As written in the original manuscript, we compared our estimates against very recently published sets of new estimates (Lakhani, Tierney et al. 2019), which were used in neither training nor validation in our analysis. In our revision, we revisited the comparison by selecting only the results for a list of acute and chronic diseases, respectively (see Supplemental Table S9). We first computed absolute errors (*i.e.*, the absolute difference between model-inferred and published values), and then used the Wilcoxon rank sum test to determine whether the distribution of the errors seen in acute diseases is different from that of chronic diseases. This difference proves insignificant (p -value = 0.18, as shown in Supplemental Figure S3C), which suggests that the accuracy of model predictions for acute diseases is similar to that for chronic diseases. We therefore confirm that, as far as the proposed model is concerned, diseases, acute or chronic, are no different. They both very much benefit from the rich information contained in disease trends and comorbidity patterns, making dissecting the genetic and environmental determinants of their pathogenesis possible.

We have added these details to the Discussion section.

g) Was the uncertainty (or errors) in the estimates of prevalence used in the models as well? Or just the point estimates?

We used the expected prevalence (point estimates) value curves as features in our model building.

References

- Lakhani, C. M., B. T. Tierney, A. K. Manrai, J. Yang, P. M. Visscher and C. J. Patel (2019). "Repurposing large health insurance claims data to estimate genetic and environmental contributions in 560 phenotypes." Nat Genet **51**(2): 327-334.
- Tylee, D. S., J. Sun, J. L. Hess, M. A. Tahir, E. Sharma, R. Malik, B. B. Worrall, A. J. Levine, J. J. Martinson, S. Nejentsev, D. Speed, A. Fischer, E. Mick, B. R. Walker, A. Crawford, S. F. A. Grant, C. Polychronakos, J. P. Bradfield, P. M. A. Sleiman, H. Hakonarson, E. Ellinghaus, J. T. Elder, L. C. Tsoi, R. C. Trembath, J. N. Barker, A. Franke, A. Dehghan, T. Me Research, C. C. Inflammation Working Group of the, M. C. o. t. I. S. G. Consortium, R. Netherlands Twin, C. W. G. neuro, C. Obsessive, C. Tourette Syndrome Working Group of the Psychiatric Genomics, S. V. Faraone and S. J. Glatt (2018). "Genetic correlations among psychiatric and immune-related phenotypes based on genome-wide association data." Am J Med Genet B Neuropsychiatr Genet **177**(7): 641-657.
- Visscher, P. M., W. G. Hill and N. R. Wray (2008). "Heritability in the genomics era--concepts and misconceptions." Nat Rev Genet **9**(4): 255-266.

Reviewer #2 (Remarks to the Author):

NCOMMS-19-12796

Estimating Genetic Parameters in the Absence of Genetic Data from Country-scale Health Datasets

This paper by Jia and colleagues has two major contributions. First, using large health care datasets, the authors computed disease prevalence curves and low dimensional disease embeddings which revealed similarity and dissimilarity of temporal disease trajectories across the disease spectrum. Second, the authors showed that disease-specific features and disease-disease relationships can be used to imputed unseen disease heritability and genetic correlation measures with high accuracy. Overall I think this is an interesting paper with many novel ideas and I applaud the authors for putting together very large scale health care datasets across countries.

However, I'm disappointed about the writing of this paper. I think it was initially written as a short letter to target a different journal. As a result, introduction, results and discussion were lumped together; the background was not thoroughly reviewed and motivation wasn't clear; and the implications of the results were not discussed. More importantly, the methods section is too brief to provide enough details to replicate the analysis. With the level of methodological details presented it's difficult to assess the rigor of the methods. In addition, the authors wanted to show lots of analyses and results in this paper but the flow of the text was sort of poor, making it difficult to follow. One has to jump between the main text, methods and figure legends to figure out what has been done.

If possible, I would suggest the authors rewrite and resubmit the paper. The paper is interesting but needs to be expanded and polished to more clearly present the results and methods, and to add background introduction and discussion.

Thank you for your feedback and thoughtful comments, we did our best to carefully revise our manuscript to meet this reviewer's requests.

Reviewers' Comments:

Reviewer #1:

Remarks to the Author:

The authors have significantly revised the manuscript according to my feedback. Overall, I think the revision is acceptable. I have a few additional minor comments.

1. I would like to see the relative feature performances of the embeddings, the disease onset data, and the other additional covariates. I am curious whether one type of data is accounting for the majority of the performance at predicting h_2 and $dcorr$. I didn't see it mentioned or a figure in the supplement (although I may have missed it).
2. I think it's probably worth noting that the author's "fourth approach" is only made possible by the existence of the other three approaches.
3. Figure 2 seems like an overly complicated visualization to me for the data being shown. I'm not sure the 3D shading or disks around the spheres are really adding anything. To paraphrase Edward Tufte, any visual element that doesn't convey information should be removed.

Reviewer #2:

Remarks to the Author:

NCOMMS-19-12796A

Estimating genetic parameters in the absence of genetic data from country-scale health datasets
Jia et al.

The authors have significantly improved the paper, although overall I think the main text can be further polished to improve readability and clarity. I have some detailed suggestions below, most of them are very minor.

- * title: I'd suggest the author replace "genetic parameters" with "heritability and genetic correlations", as it's unclear what do genetic parameters refer to.
- * Figure 1: Can the authors clarify the dataset used to compute prevalence curve similarity and perform clustering analysis in Figure 1B? Did the authors use the US data only or combine multiple datasets?
- * As disease curves have been shifted along the x-axis, does it still make sense to talk about early vs. late-raising curves in Figure 1C?
- * In the main text, the predicted genetic parameters were validated using a 4:1 (training vs testing) cross-validation with 1000 random splits, but in the caption of Figure 3 the description was inconsistent (a 2:1 cross-validation and 10000 random splits).
- * It's difficult to make sense of the numbers on the density plots (Figure 3B-C). Would it be possible to normalize these numbers and make them more interpretable?
- * In the results section "Building estimators from disease descriptors", it'd be helpful to briefly describe the prediction model and predictors (features and covariates) to improve the flow of the text. Otherwise it's a little difficult to follow without looking into the methods section. It's also helpful in this section to always report the datasets (US or US & Denmark combined etc) used and the number of data points to compute the Pearson correlations.
- * In the paragraph starting on line 217, it's unclear what did the authors do when there are more

than two previous published estimates for the same disease.

* Line 422: what is the rationale of defining "disease onset age" as the maximum age among 5% of youngest patients carrying the disease, as opposed to something like the average age of getting a particular disease code for the first time?

Reviewers' comments (in black) and proposed answers and solutions (in red):

We thank both reviewers for their insightful suggestions. As a result of the revision the manuscript has been improved significantly. We are glad to polish it further and carefully according to the reviewers' feedbacks below. All the new changes in the manuscript text file are highlighted in yellow.

Reviewer #1 (Remarks to the Author):

The authors have significantly revised the manuscript according to my feedback. Overall, I think the revision is acceptable. I have a few additional minor comments.

1. I would like to see the relative feature performances of the embeddings, the disease onset data, and the other additional covariates. I am curious whether one type of data is accounting for the majority of the performance at predicting h_2 and d_{corr} . I didn't see it mentioned or a figure in the supplemented (although I may have missed it).

Thank you for the constructive suggestion. In Table 2 that reports the feature importance, we now have added a detailed breakdown of 20 embedding factors as well as other covariates, such as disease onset age, disease category, country of cohort, and sex of cohort used. We can see that the 20 embedding factors are all important, and the proportions of their contributions towards model prediction are similar.

2. I think it's probably worth nothing that the author's "fourth approach" is only made possible by the existence of the other three approaches.

We have inserted the following wording in the introduction section:

"These accumulating legacy estimates of genetic parameters, such as heritability and genetic correlations, paved way for the fourth approach that we are proposing here."

3. Figure 2 seems like an overly complicated visualization to me for the data being shown. I'm not sure the 3D shading or disks around the spheres are really adding anything. To paraphrase Edward Tufte, any visual element that doesn't convey information should be removed.

We re-generated Figure 2 without distracting discs, as suggested by the reviewer.

Reviewer #2 (Remarks to the Author):

NCOMMS-19-12796A

Estimating genetic parameters in the absence of genetic data from country-scale health datasets

Jia et al.

The authors have significantly improved the paper, although overall I think the main text can be further polished to improve readability and clarity. I have some detailed suggestions below, most of them are very minor.

* title: I'd suggest the author replace "genetic parameters" with "heritability and genetic correlations", as it's unclear what do genetic parameters refer to.

As suggested, the new title now is "Estimating Heritability and Genetic Correlations from Country-scale Health Datasets in the Absence of Genetic Data"

* Figure 1: Can the authors clarify the dataset used to compute prevalence curve similarity and perform clustering analysis in Figure 1B? Did the authors use the US data only or combine multiple datasets?

For curve similarity computation and clustering analysis, we used datasets from two countries, *i.e.*, US and Denmark, to:

- (1) pinpoint whether certain curve patterns are country-specific or consistent between the two countries;
- (2) analyse how country-specificity affects clustering (in Figure 1C, the third stacked bar chart summarizes the country compositions in each cluster, and for example, we saw US-based disease curves made up a larger proportion in Cluster 3, while the proportions of Denmark-based ones were higher in Clusters 4 and 5).

We have now explained explicitly in Methods 2 (*Clustering disease prevalence curve shapes*) that the clustering analysis was based on US and Denmark data.

* As disease curves have been shifted along the x-axis, does it still make sense to talk about early vs. late-rising curves in Figure 1C?

Thank you for bringing this question up. The relative shift between disease curves was local, *i.e.*, -8 to +8 years, and this shift would not alter the global, 65-year course of a disease, especially in terms of the slope and multimodality. One claim we made on Figure 1C that "Clusters 2 and 4 include early- and later-rising reversed L-shaped curves, respectively" is really about their slope differences. The curves in Cluster 2, if fitted with a linear line and compared against the curves in Cluster 4, have more gentle slopes and intersect the Age-axis much earlier, suggesting an earlier-rising trend.

In the manuscript, we have changed the statement “Clusters 2 and 4 include early- and later-rising reversed L-shaped curves, respectively” to “Clusters 2 and 4 include reversed L-shaped curves (the former being early- but slow-rising, while the latter being later- but steeper-rising)”, in order to emphasize the global trend differences between the curves in Clusters 2 and 4.

* In the main text, the predicted genetic parameters were validated using a 4:1 (training vs testing) cross-validation with 1000 random splits, but in the caption of Figure 3 the description was inconsistent (a 2:1 cross-validation and 10000 random splits).

Thank you for pointing this out. To be consistent, we now use the same settings, *i.e.*, a 4:1 (training vs testing) cross-validation with 1000 random splits to regenerate Figure 3B-C. Our claim that “the slopes of both linear regressions were close to 1, with negligible intercepts, indicating that our estimates were nearly perfectly unbiased” still holds.

* It's difficult to make sense of the numbers on the density plots (Figure 3B-C). Would it be possible to normalize these numbers and make them more interpretable?

We admit that there lacks an interpretation about the density plots, and thus propose the following as remedies.

The reason why we chose the density plots has now been added in the legends of Figure 3B-C as:

“We used these plots to visualize the joint distribution of the actual data for testing and model-predicted values. “

The interpretation about the results has now been inserted in the main text as follows:

“Contour plots in Figure 3B-C show our model predictions’ estimated densities against published estimates of corresponding parameters: in the case of h^2 , the density peaked around (0, 0) and (0.4, 0.4), indicating denser collocations of published and predicted estimates there; while as for *corr*, the estimates exhibited a unimodal distribution with a peak close to (0.05, 0.05).”

The numbers on the plots indicate the density estimates, and by default, the kernel density estimates were already normalized. We hope the added explanations above would make the density plots more interpretable.

* In the results section "Building estimators from disease descriptors", it'd be helpful to briefly describe the prediction model and predictors (features and covariates) to improve the flow of the text. Otherwise it's a little difficult to follow without looking into the methods section. It's also helpful in this section to

always report the datasets (US or US & Denmark combined etc) used and the number of data points to compute the Pearson correlations.

Thank you for the constructive suggestion. We have added the following snippet at the beginning of the respective results section:

“Disease prevalence curves and disease embedding derived from the US dataset were used as disease-specific descriptors for modeling. The modeling features also included specifications about predicted estimates (data type and mathematical model used), basic information about the investigated cohorts (country of origin and sex), and disease characteristics (category of biological systems that the disease belongs to, and the onset age). A detailed description of disease features used in the model can be found in Methods, part 4.”

We have now indicated the numbers of used data points in brackets for the computation of the Pearson correlations.

* In the paragraph starting on line 217, it's unclear what did the authors do when there are more than two previous published estimates for the same disease.

“If there were more than two estimates of the same type, we used all of them, generating all possible comparison pairs.”

We made this clarification in the revised manuscript by inserting the specification above in the paragraph starting on line 217.

* Line 422: what is the rationale of defining "disease onset age" as the maximum age among 5% of youngest patients carrying the disease, as opposed to something like the average age of getting a particular disease code for the first time?

Due to the limited time window at which we can observe individuals in the US datasets, the actual age of onset for each patient is unknown. Hence, we defined an upper-bound of the earliest disease age onset statistically, as the age separating the youngest five percent of the disease carriers from the rest. The results would remain qualitatively the same if one percent or ten percent cutoff was chosen.

Reviewers' Comments:

Reviewer #1:

Remarks to the Author:

I am satisfied with this revision.

Reviewer #2:

Remarks to the Author:

All my comments have been addressed, and I think the current version of the paper is acceptable. That said, I'd suggest the authors proofread carefully- there are still quite a few typos and grammatical errors in the text. In fact, I still think the overall writing quality can be improved and another round of editing for clarity would be helpful.

REVIEWERS' COMMENTS:

Reviewer #1 (Remarks to the Author):

I am satisfied with this revision.

Thank you.

Reviewer #2 (Remarks to the Author):

All my comments have been addressed, and I think the current version of the paper is acceptable. That said, I'd suggest the authors proofread carefully- there are still quite a few typos and grammatical errors in the text. In fact, I still think the overall writing quality can be improved and another round of editing for clarity would be helpful.

Thank you for the suggestion. We re-visited the manuscript carefully to eliminate typos and to improve the writing quality.